# Earlier Age at Surgery for Brain Cavernous Angioma-Related Epilepsy May Achieve Complete Seizure Freedom without Aid of Anti-Seizure Medication

**DOI:** 10.3390/brainsci12030403

**Published:** 2022-03-18

**Authors:** Ayataka Fujimoto, Hideo Enoki, Keisuke Hatano, Keishiro Sato, Tohru Okanishi

**Affiliations:** 1Department of Neurosurgery, Seirei Hamamatsu General Hospital, Shizuoka 430-8558, Japan; 2Comprehensive Epilepsy Center, Seirei Hamamatsu General Hospital, Shizuoka 430-8558, Japan; enokih.neuropediatr@gmail.com (H.E.); hatakenosuke@gmail.com (K.H.); k-sato@sis.seirei.or.jp (K.S.); t.okanishi@tottori-u.ac.jp (T.O.)

**Keywords:** epilepsy, cavernous angioma, anti-seizure medication, early surgery, age at surgery, multifactorial mechanism

## Abstract

Background: The present study hypothesized that some factors may distinguish between patients with a brain cavernous angioma (BCA), who were free from anti-seizure medication (ASM), and patients who still required ASMs postoperatively. The purpose of the study was thus to identify factors associated with ceasing ASMs for patients with drug-resistant epilepsy secondary to BCA, who underwent BCA removal surgery. Methods: We divided patients into those with drug-resistant epilepsy secondary to BCA who achieved complete seizure freedom without ASMs a year after surgery (No-ASM group) (International League Against Epilepsy (ILAE) classification class I with no epileptiform discharges), and others (ASM group) (ILAE classification ≤ II and/or epileptiform discharges). We statistically compared groups in terms of: (1) age at operation; (2) history of epilepsy; (3) size of BCA; and (4) location of BCA. Results: Overall, a year after the surgery, the No-ASM group comprised 12 patients (48%), and the ASM group comprised 13 patients (52%). In both multi- and univariate logistic regression analyses, age at BCA removal surgery correlated significantly with the No-ASM group (*p* = 0.043, *p* = 0.019), but history of epilepsy did not (*p* = 0.581, *p* = 0.585). Conclusions: Earlier age at surgery for patients with drug-resistant epilepsy is encouraged to achieve complete seizure freedom without the need for ASMs when the cause of epilepsy is BCA.

## 1. Introduction

A brain cavernous angioma (BCA) is a benign, non-arterial, angiomatous lesion that has thin and sinusoidal blood vessel walls containing endothelium and collagenous adventitia surrounded by hemosiderin-storing macrophages and gliosis without brain parenchyma. BCAs may show calcification or ossification [1,2].

Some studies have shown BCAs in 0.4–0.7% of the population [3,4]. BCAs are also known as cavernomas, cavernous malformations, and cavernous hemangiomas. About 85% of BCAs are located supratentorially, while the rest are infratentorial [5,6,7]. About 15% of BCAs occur as familial (hereditary) forms, but the vast majority are sporadic [3,8,9]. In the hereditary form, multiple lesions are the norm [10], and are inherited in an autosomal-dominant pattern, typically with known gene mutations [11,12,13,14]. In the sporadic form, single or few lesions predominate. BCAs are symptomatic in 15–80% of cases [3,15,16].

Seizures are seen in 10–70% of patients with a BCA, headaches in 30%, neurological manifestations including paresis in 40–50%, and bleeding in 20% [17,18,19].

Among BCA patients with epilepsy, 70% are drug-resistant [20]. For drug-resistant epilepsy secondary to BCA, surgical treatment has been established [21,22], and the rate of freedom from seizures after surgery ranges from 60% to more than 90% [19,23,24]. About 70–80% patients who undergo BCA-removal surgery can discontinue anti-seizure medications (ASMs) [25,26]. However, whether surgical treatment can achieve freedom from seizures without ASMs, and what factors are related to the cessation of ASMs, remain unclear. Even though surgical treatment can achieve a high rate of freedom from seizures, seizure freedom that requires the assistance of ASMs is not a radical cure. Evaluation of the requirement for ASMs after BCA removal might therefore be important in patients with drug-resistant epilepsy secondary to BCA.

For BCAs, whether hemosiderin deposits should be removed or a pure lesionectomy at the time of surgery is sufficient, was an issue that was widely discussed in the late 1980s to 1990s [27,28], but the International League Against Epilepsy (ILAE) concluded that the epileptogenicity of a BCA is not related to hemosiderin deposition alone, but instead involves multiple mechanisms [29,30].

However, factors related to whether freedom from ASM use can be achieved after surgery have rarely been investigated for drug-resistant epilepsy secondary to BCA [25,31,32]. This study hypothesized that some factors differ between patients with BCAs who are free from ASM use postoperatively, and those who still require ASMs. The purpose of this study was thus to identify factors associated with ending the use of ASMs for patients with drug-resistant epilepsy secondary to BCAs who underwent BCA removal.

## 2. Methods

### 2.1. Study Design and Ethics Approval

The ethics committee at Seirei Hamamatsu General Hospital approved the protocol for this study (approval no. 3857), which was performed in accordance with the principles of the Declaration of Helsinki. All participants provided written informed consent prior to inclusion in the study.

This was a retrospectively reviewed, observational study comparing two groups. We performed surgery or adjustment of ASMs for the purposes of treatment, but not for the purposes of this study.

### 2.2. Clinical Information

We retrospectively reviewed our electronic medical records and identified patients with BCAs who underwent BCA removal between August 2009 and August 2020.

Inclusion criteria were: (1) patients with epilepsy secondary to supratentorial BCA; (2) patients with intractable epilepsy who received more than three ASMs for a total of more than two years [33]; (3) patients who underwent BCA removal surgery; and (4) patients with a postoperative follow-up >1 year.

Patients without epilepsy who only underwent removal surgery due to mass effect or hemorrhage from the BCA were excluded from this study. Patients who underwent BCA removal and even achieved seizure freedom without epileptiform discharges on scalp electroencephalography (EEG), but still received ASMs because of personal reasons (e.g., anxiety, drivers’ license issues, etc.), were also excluded from this study.

### 2.3. Magnetic Resonance Imaging of the Brain

All patients underwent 3-T magnetic resonance imaging (MRI) of the brain (Discovery MR-750; GE Inc., Chicago, IL, USA). To identify and count BCAs, we used T2*-weighted, T2-fast imaging employing steady-state acquisition, and fluid-attenuated inversion-recovery (FLAIR) sequences. To determine the size of the BCA, we measured the longest diameter of the solid part of the BCA, not the entire diameter of the hemosiderin ring on FLAIR imaging. In the case of multiple BCAs, we measured the size of the BCA most responsible for epileptogenicity that was surgically removed.

### 2.4. Surgery and Surgical Outcome

We performed a transcortical BCA removal. The shortest trajectory from the surface of the brain to the BCA was taken. At the surface of the BCA, adjacent brain tissue that included hemosiderosis was removed by an ultrasonic aspirator to isolate and remove the BCA. However, we did not completely aspirate hemosiderin deposits that could be identified at the time of surgery. Specimens were submitted for histopathological studies.

We evaluated postoperative epilepsy outcomes at 6 and 12 months postoperatively using established ILAE criteria [34]. In addition, postoperative seizure outcomes used for this study were recorded at the time of last follow-up evaluation at an outpatient clinic for patients with more than 1 year of follow-up.

### 2.5. Follow-Up Evaluation

Patients underwent scalp EEG 6 months after surgery using the international 10–20 system. Patients were tapered off ASMs if they: (1) showed no clinical seizures (ILAE classification class I; complete freedom from seizures, no aura); (2) displayed complete lesionectomy without hemosiderin deposits on follow-up MRI; (3) showed no epileptiform discharges on scalp EEG (based on a previous study that showed a correlation between postoperative seizure recurrence and epileptiform discharge [35]; and (4) were willing to quit ASMs. However, patients were required to remain on at least one ASM if they: (A) experienced clinical seizures including aura; or (B) showed epileptiform discharges on scalp EEG; or both A and B. All patients underwent EEG and evaluation of seizure outcome 1 year postoperatively. EEGs were evaluated by certified epileptologists (A.F., K.S.).

At this time, we divided patients into two groups: a group with ILAE classification class I (complete freedom from seizures, no aura), without epileptiform discharges on scalp EEG, who were not on ASMs (No-ASM group); and a group with ILAE class ≤ II epileptiform discharges on EEG, or both, who were on ASMs (ASM group) (Figure 1).

### 2.6. Primary Outcome Measurement

We divided patients who achieved complete seizure freedom without ASMs at 1 year after surgery (ILAE classification class I with no epileptiform discharges on scalp EEG) from others (ILAE classification ≤ II and/or epileptiform discharges on scalp EEG). We statistically compared these two groups in terms of: (1) age at operation; (2) history of epilepsy; (3) size of BCAs; and (4) location of BCAs. We did not compare single and multiple lesions because only 2 patients showed multiple lesions. As sex was almost evenly balanced between groups, this factor was also excluded from analysis.

### 2.7. Secondary Outcome Measurement

Surgical complications and number of pre- and postoperative ASMs were reviewed. As only one patient had a BCA in the occipital area, and all others had BCAs in the frontal or temporal regions, we also analyzed patients with temporal- and frontal-lobe BCAs, excluding the single patient with BCA in the occipital area.

### 2.8. Statistical Analysis

The Mann–Whitney U-test and Student’s *t*-test were used in statistical analyses of differences in clinical data between the No-ASM and ASM groups, as appropriate.

Comparisons between these two groups were also made using multivariate logistic regression tests. Values of *p* < 0.05 were considered to indicate significant differences in all analyses. All statistical analyses were performed using Sigma Plot version 14.5 software (Systat Software, San Jose, CA, USA).

## 3. Results

### 3.1. Clinical Information

Seventy patients with BCAs were identified from the records. Among these, 25 patients (18 males, 7 females) met the study criteria. Ages ranged from 9 to 71 years (mean, 36.4 years; median, 35 years; standard deviation, 16.5 years). All patients underwent BCA removal surgery and postoperative follow-up ranged from 1 year to 18 years (mean, 6.7 years; median, 6.5 years; standard deviation, 4.2 years) (Table 1).

### 3.2. Primary Outcome

Fourteen patients: (1) achieved complete seizure freedom (ILAE classification class I); (2) showed no scalp EEG epileptiform discharges; (3) showed complete lesionectomy without hemosiderin deposits as evaluated by follow-up MRI; and (4) were able to quit ASMs 6 months after removal surgery.

Eleven patients did not satisfy these four conditions by 6 months postoperatively (1 patient with clinical seizures, 10 patients with epileptiform discharges).

Two more patients showed recurrence of clinical seizures during the following 6-month period until the 1-year postoperative evaluation and were thus categorized into the ASM group at 1 year after surgery.

Overall, 12 patients satisfied the conditions for assignment to the No-ASM group (48%), and 13 patients (52%; 3 patients with clinical seizures, 10 patients with epileptiform discharges without clinical seizures) were assigned to the ASM group at 1 year postoperatively (Figure 1). Patients in the No-ASM and ASM groups maintained those conditions until the latest visit of the follow-up period. A MRI at 6 months postoperatively revealed that all participants obtained complete lesionectomy and removal or absorption of surrounding hemosiderin deposits.

Univariate logistic regression analyses showed that age at the time of BCA removal correlated significantly with group allocation (*p* = 0.019). Multivariate logistic regression analysis also showed that age at the time of BCA removal surgery correlated significantly with group allocation (*p* = 0.043), while epilepsy history did not (*p* = 0.581) (Table 2).

### 3.3. Secondary Outcomes

No surgical complications were encountered in this study. While the number of ASMs prescribed did not differ significantly between groups (*p* = 0.28), the number of ASMs administered postoperatively was, unsurprisingly, significantly lower in the ASM group (*p* < 0.001).

In the ASM group, one patient still suffered from aura, but no others exhibited clinical seizures at the latest visit. The postoperative seizure-free rate without the aid of ASMs was 48% (12 patients), whereas the seizure-free rate with and without the aid of ASMs was 96% (24 patients) at the time of the latest visit in this study.

Only one patient showed a BCA in the occipital area. Excluding that case, when patients were separated into those showing either frontal- or temporal-lobe BCAs, age at surgery remained significantly correlated with use of ASMs (*p* = 0.046) (Table 3).

## 4. Discussion

In the present study, the main factor associated with whether freedom from ASM use could be achieved was age at surgery, and ASMs were more likely to be unnecessary if surgery was performed at a younger age, such as young adulthood. Since the duration of the history of epilepsy did not differ significantly, this difference might reflect the difference in age at epilepsy onset.

De novo BCAs can reportedly be induced by cerebral radiation [36,37], but the concept of a BCA as a congenital lesion is generally accepted [4].

Among patients with similar MRI lesions, locations, and background characteristics (age, sex, etc.), our clinical experience indicates that one may have ASM-resistant epilepsy, another may have ASM-controllable epilepsy, and another may be asymptomatic. Even with familial BCAs, some patients may be asymptomatic and some may be symptomatic within the same family. Differences may thus exist between brains, with or without epileptogenicity secondary to BCA.

One old and well-known notion regarding the mechanisms underlying the epileptogenicity of BCAs is hemosiderin [38]. In addition to hemosiderin, imbalances between excitatory and inhibitory neurotransmitters have been suggested to be influenced by blood components such as albumin [39], and might contribute to the epileptogenicity of BCAs [40]. If the mechanism underlying the epileptogenicity of BCAs is simply related to the single-hit theory of the leakage of components from the BCA, pure lesionectomy would represent a gold standard. However, the fact that pure lesionectomy cannot always achieve complete freedom from seizures means that the epileptogenicity of BCAs is not so simple to define, and potentially involves complex and multifactorial pathways [41,42]. Therefore, in terms of multifactorial mechanisms, epileptogenicity as a result of interactions between BCAs and other factors, such as focal cortical dysplasia (FCD) [43] and amyloid beta [43], has been suggested. Given this background, BCA-related epilepsy also seems to be a network disease [44,45,46,47]. The greater the number of factors that the acquired epileptogenicity of a BCA involves, the more complicated the mechanisms involved in the epilepsy network of FCD, amyloid beta, etc.

Many papers have recommended early surgery for patients with epilepsy secondary to BCAs [48,49,50,51], whereas just one paper reported better outcomes from later surgery [23]. The present results may support early surgery for patients with drug-resistant epilepsy to achieve complete freedom from seizures without the need for ASMs when the cause of epilepsy is a BCA.

However, a key limitation of this study is that we did not examine whether the shorter history of epilepsy seen with early surgery was associated with increased ASM-free rates in the ASM group. Future work should examine age-matched groups with different durations of epilepsy history to evaluate postoperative rates of complete seizure freedom without ASMs.

In addition to these limitations, the location of BCAs showed a relatively low *p*-value. Since patients can quit ASMs based on the location of the BCA, which could therefore represent important preoperative information, a study with a larger cohort is also warranted. Little is known regarding long-term outcomes or the relevance of BCA location to BCA-related epilepsy due to the limited number of patients in this study, but this preliminary study provides a valuable starting point for future research.

## 5. Conclusions

Earlier age at surgery for patients with drug-resistant epilepsy is encouraged to achieve complete freedom from seizures without the need for ASMs when the cause of epilepsy is BCA.

## Figures and Tables

**Figure 1 brainsci-12-00403-f001:**
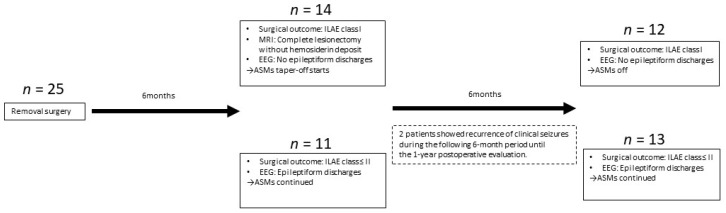
Patient assignment to No-ASM and ASM groups. Overall, 12 patients satisfied the conditions for inclusion in the no-anti-seizure medication (ASM) group (48%), and 13 patients (52%) (3 patients with clinical seizures and 10 patients with epileptiform discharges without clinical seizures) were assigned to the ASM group a year after surgery.

**Table 1 brainsci-12-00403-t001:** Clinical information.

Seizuire Freedom	ILAE Class I (Complete Seizure Freedome) without Epileptiform Discharges	ILAE Class ≤II and/or Epileptiform discharges	*p*-Value
Number of patients	12(48%)	13(52%)	n/a
Sex (Female: Male)	3:9	4:9	0.75
Age at surgery (mean, range, SD)	28.6, 10–45, 11.1	43.6, 9–62, 17.7	0.018 *
Epilepsy history (year) [mean, range, SD]	9.2, 2–31, 9.1	11.5, 2–35, 12.0	0.64
Single vs. Multiple lesions (Single: Multiple)	12:0	11:2	0.34
Pre-perative ASM (mean, range, SD)	2.4, 1–4, 1.2	2.9, 1–5, 1.0	0.28
Number of Post-operative ASM (mean, range, SD)	n/a	1.5, 1–3, 0.7	n/a
Size of BCA (mm) [mean, range, SD]	28.4,1.5–190, 51.8	8.9, 5–16.1, 3.3	0.15
Post operative follow-up period (year)	7.3, 1–18, 4.9	6.2, 1–12, 3.5	0.55
Location of BCA			0.08
Frontal	4	1	
Temporal	8	11	
Occipital	0	1	

ASM: anti-seizure medication; BCA: brain cavernous angioma; ILAE: international league against epilepsy; n/a: not available; SD: standard deviation; Statistics: progressive myoclonic epilepsy; * Statistically significant.

**Table 2 brainsci-12-00403-t002:** Statistic Analyses of predictors for ILAE class I without epileptiform discharges on scalp EEG.

	Coefficient	Standard Error	*p*-Value
Multi-variate logistic regression analysis			
Age at surgery (year old)	0.133	0.066	0.043 *
Size of BCA	−0.072	0.124	0.562
Epilepsy history (years)	−0.028	0.051	0.581
Location of BCA	3.204	2.122	0.131
Univariate-logistic regression analysis			
Age at surgery (year old)	0.0144	0.006	0.019 *
Size of BCA	−0.004	0.003	0.191
Epilepsy history (years)	0.006	0.01	0.585
Location of BCA	0.296	0.166	0.088

BCA: brain cavernous angioma; EEG: electroencephalography; ILAE: International League Against Epilepsy; *: significant correlations to the complete seizure freedom (ILAE class I) without epileptiform discharge on EEG; Statistics: multivariate logistic regression analysis.

**Table 3 brainsci-12-00403-t003:** Statistical analyses of predictors for ILAE class I without epileptiform on scalp EEG.

	Coefficient	Standard Error	*p*-Value
Multi-variate logistic regression analysis			
Age at surgery (year old)	0.130	0.065	0.046 *
Size of BCA	−0.075	0.123	0.543
Epilepsy history (years)	−0.026	0.051	0.61
Location of BCA	2.787	2.497	0.264

BCA: brain cavernous angioma; EEG: electroencephalography; ILAE: International League Against Epilepsy; *: significant correlations to the complete seizure freedom (ILAE class I) without epileptiform discharge on EEG; Statistics: multivariate logistic regression analysis.

## Data Availability

Data supporting the findings of this study are available from the first author (A.F.) upon reasonable request.

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
