# Peer review of "Earlier Age at Surgery for Brain Cavernous Angioma-Related Epilepsy May Achieve Complete Seizure Freedom without Aid of Anti-Seizure Medication"

_brainsci, 2022, doi:10.3390/brainsci12030403_

Round 1

Reviewer 1 Report

Although the number of subjects and thus statistical power are small, this study does add meaningfully to the body of knowledge on the possible factors involved in the success of surgery to end seizures due to BCM.

Introduction, paragraph 2, last sentence: 15% of what? Change to "15% to 80% of BCM is symptomatic."

Introduction, paragraph 3: Change to "10-70% of patients, headaches in 30%, ...".

Introduction, paragraph 4: Change to "Among BCM patients with epilepsy, 70% are drug-resistant [19]."

Introduction, paragraph 4: Instead of telling us the maximum rate of freedom from seizure ("can exceed 90%"), tell us the expected rate, or average. Give more information from the literature. Does the freedom from seizure still require drugs or not? What fraction of patients after surgery need drugs or don't need drugs? Or are there no reports in the literature?

Introduction, paragraph 4: Why "Therefore"? You haven't explained why this is unclear. Are there no reports on this in the literature? If so, then say so. Are there published reports on what factors are related to cure without ASMs? What does "radical" mean in this context?

Section 2.3: Would it be simpler to say "we measured the longest diameter of the solid part of the BCA, but..."?

Section 2.4: Could "gross totally identified" be written more clearly? 

Section 2.5: In the section title, why say "follow-up" only for MRI, but not for Anti-seizure medication or for EEG? They are also follow-ups. You might change the title to something like "Follow-up evaluation". You may move the 2nd paragraph of section 2.4 to section 2.5.

Section 2.5: Rewrite this to clarify what you mean: ", However, patients who had clinical seizures including aura, or showed epi-leptiform discharges on scalp EEG or both were potentially able to reduce ASMs, but were required to remain on at least one ASM."

Figure 1: Enlarge the font to make it easier to read. What is "6m"? If it's 6 months, then say 6 months. For the 2 patients, you may show them as a horizontal arrow from left to right. You may delete the latest visits since they don't add useful information.

Section 2.8: 4 of the 6 variables you analyzed are categorical, so you should use chi square tests on them, not Mann-Whitney or Student's t tests.

I wonder whether the location of BCA in the gray matter vs. white matter makes any difference in the outcome. Have you looked at this variable in your data?

Table 1: Change to "Pre-operative ASM".

Section 3.2, paragraph 5: Change p = 0.040 to p = 0.038 to match Table 2, if that's correct. Or you can say p = 0.04, without the extra zero in the thousandths place.

Section 3.2, paragraph 6: Correct the grammar.

Section 3.2, paragraph 6: There's only a single case with an occipital lobe BA. What if you lump it with either the temporal or the frontal cases? How would the p-value look then?

Table 2: There are only 2 cases with multiple lesions, and there's no way that 2 cases would give a significant association, so you should remove "Single lesion" from the logistic regression analysis. Sex is also almost exactly evenly balanced between the two groups, so perhaps you should remove it as well. There's only 1 case located outside either the frontal or temporal lobes, so it would make sense to lump it with either frontal or temporal cases (to avoid weak power, and also to deal with the problem that the arbitrary numbers you assign to each location [1 for frontal, 2 for temporal, and 3 for occipital? Or some other arbitrary coding system?] will affect the results of the regression, which wouldn't make sense. But if there are only 2 locations, then the arbitrary value you assign to each one becomes irrelevant, which solves the problem of the arbitrariness.)

What if you analyze age at surgery on its own? Do the age distributions satisfy prerequisites for doing a t test (normal distributions with similar SDs)? If so, what is the p-value of the t test? If not, what is the p-value of the U test? You may show a scatter plot of age at surgery for the two groups so that we can see the difference in distributions.

In Section 2.8, you mentioned U-tests and t-tests, but you don't show results of these; why not?

Section 3.3: How were these analyzed? They're not in Table 2.

Discussion, penultimate paragraph: Why do many papers recommend early surgery? This seems to contradict your statement, in the last paragraph of the Introduction, that factors related to freedom from ASM use have rarely been investigated. You may include some of the results of these papers showing that early surgery gives better outcomes.

Discussion, penultimate paragraph: How do your results support early surgery? Your analysis shows no association of epilepsy duration with seizure freedom, p=1.0.

Discussion, last paragraph, 1st sentence: Didn't you examine this? See above.

Discussion, last paragraph: Change "examined" to "examine".

Discussion, last paragraph: Why might some patients in the ASM group be categorized to the No-ASM group if follow-up had been extended? I don't understand.

Author Response

Responses to Comments from Reviewer 1

Thank you very much for reviewing our manuscript. We greatly appreciate the advice provided.

We are having the revised version of the manuscript checked by a professional English-speaking editor. However, the return from the proofreading service will take several more days, which would take this resubmission past the deadline. Please understand that this revised version of the manuscript is therefore provided “before external proofreading”.

Although the number of subjects and thus statistical power are small, this study does add meaningfully to the body of knowledge on the possible factors involved in the success of surgery to end seizures due to BCM.

Introduction, paragraph 2, last sentence: 15% of what? Change to "15% to 80% of BCM is symptomatic."

Response: Thank you very much for this advice. We have changed the phrasing based on this suggestion.

Introduction, paragraph 3: Change to "10-70% of patients, headaches in 30%, ...".

Response: Thank you very much for this advice. We have changed the text accordingly.

Introduction, paragraph 4: Change to "Among BCM patients with epilepsy, 70% are drug-resistant [19]."

Response: Thank you very much for this advice. We have amended this sentence as suggested.

Introduction, paragraph 4: Instead of telling us the maximum rate of freedom from seizure ("can exceed 90%"), tell us the expected rate, or average. Give more information from the literature. Does the freedom from seizure still require drugs or not? What fraction of patients after surgery need drugs or don't need drugs? Or are there no reports in the literature?

Response: Thank you very much for this advice. We have amended the text to show that the range was 60–90%.

Introduction, paragraph 4: Why "Therefore"? You haven't explained why this is unclear. Response: We have rephrased this part.

Are there no reports on this in the literature? If so, then say so. Are there published reports on what factors are related to cure without ASMs?

Response: Since no reports have been published regarding the discontinuation of ASM after removal surgery, we have mentioned this information in the revised text.

What does "radical" mean in this context?

Response: We are sorry for the unclear expression. We have clarified this in the revised version of the text.

Section 2.3: Would it be simpler to say "we measured the longest diameter of the solid part of the BCA, but..."?

Response: We have rephrased this part as suggested.

Section 2.4: Could "gross totally identified" be written more clearly? 

Response: We agree that the words “gross total” were unclear. We have removed this phrase.

Section 2.5: In the section title, why say "follow-up" only for MRI, but not for Anti-seizure medication or for EEG? They are also follow-ups. You might change the title to something like "Follow-up evaluation".

Response: We have changed the title accordingly.

You may move the 2nd paragraph of section 2.4 to section 2.5.

Response: Thank you very much for this thoughtful suggestion. Since we wish to present the surgical outcomes, we would like to keep this paragraph in the original position. We hope this is acceptable.

Section 2.5: Rewrite this to clarify what you mean: ", However, patients who had clinical seizures including aura, or showed epi-leptiform discharges on scalp EEG or both were potentially able to reduce ASMs, but were required to remain on at least one ASM."

Response: We are sorry for providing an insufficient explanation. We have rewritten the phrases in question.

Figure 1: Enlarge the font to make it easier to read. What is "6m"? If it's 6 months, then say 6 months.

Response: We have increased the font size and have changed “6m” to “6 months”.

For the 2 patients, you may show them as a horizontal arrow from left to right. You may delete the latest visits since they don't add useful information.

Response: We have horizontalized the figure and have deleted the last visits.

Section 2.8: 4 of the 6 variables you analyzed are categorical, so you should use chi square tests on them, not Mann-Whitney or Student's t tests.

Response: We are sorry for not indicating the results of Mann-Whitney, and Students’ t-test in Table 1.

I wonder whether the location of BCA in the gray matter vs. white matter makes any difference in the outcome. Have you looked at this variable in your data?

Response: This is important point. However, since some BCAs involved both gray and white matter, some were not. We considered this would make the analysis complicated, so that we did not consider this factor in this study.

Table 1: Change to "Pre-operative ASM".

Response: We have changed this phrase as suggested.

Section 3.2, paragraph 5: Change p = 0.040 to p = 0.038 to match Table 2, if that's correct. Or you can say p = 0.04, without the extra zero in the thousandths place.

Response: Thank you very much for noticing this discrepancy. We have presented the data in a consistent manner.

Section 3.2, paragraph 6: Correct the grammar.

Response: We have had the grammar corrected by a professional proofreading service using native English speakers.

Section 3.2, paragraph 6: There's only a single case with an occipital lobe BA. What if you lump it with either the temporal or the frontal cases? How would the p-value look then?

Table 2: There are only 2 cases with multiple lesions, and there's no way that 2 cases would give a significant association, so you should remove "Single lesion" from the logistic regression analysis.

Sex is also almost exactly evenly balanced between the two groups, so perhaps you should remove it as well.

There's only 1 case located outside either the frontal or temporal lobes, so it would make sense to lump it with either frontal or temporal cases (to avoid weak power, and also to deal with the problem that the arbitrary numbers you assign to each location [1 for frontal, 2 for temporal, and 3 for occipital? Or some other arbitrary coding system?] will affect the results of the regression, which wouldn't make sense. But if there are only 2 locations, then the arbitrary value you assign to each one becomes irrelevant, which solves the problem of the arbitrariness.)

What if you analyze age at surgery on its own? Do the age distributions satisfy prerequisites for doing a t test (normal distributions with similar SDs)? If so, what is the p-value of the t test? If not, what is the p-value of the U test? You may show a scatter plot of age at surgery for the two groups so that we can see the difference in distributions.

Response: Thank you very much for raising these points. The patient with occipital BCA might be statistically inappropriate. We have therefore undertaken additional analysis excluding this patient and have re-analyzed the data.

Since multiple regression analysis involves each of the factors that were pointed out, simple removal of “Single lesion” and “Sex” might not be reliable. We therefore first removed these factors and then added a linear regression analysis in Table 2.

In Section 2.8, you mentioned U-tests and t-tests, but you don't show results of these; why not?

Section 3.3: How were these analyzed? They're not in Table 2.

Response: We are sorry for this confusion. We mistakenly provided Table 1 without p-values. We used U- and t-tests to compare the two groups (ASM group vs. No-ASM group). We have rephrased the parts in question and have corrected Table 1.

Discussion, penultimate paragraph: Why do many papers recommend early surgery? This seems to contradict your statement, in the last paragraph of the Introduction, that factors related to freedom from ASM use have rarely been investigated. You may include some of the results of these papers showing that early surgery gives better outcomes.

Discussion, penultimate paragraph: How do your results support early surgery? Your analysis shows no association of epilepsy duration with seizure freedom, p=1.0.

Discussion, last paragraph, 1st sentence: Didn't you examine this? See above.

Response: Thank you very much for raising these points, which we had not noticed.

Since we could not provide a recommendation for early surgery based on this study due to the lack of significance of a history of epilepsy, we have undertaken a major rephrasing of these parts as well as the conclusion.

Discussion, last paragraph: Change "examined" to "examine".

Response: We are sorry for this grammatical error. We have corrected the text accordingly.

Discussion, last paragraph: Why might some patients in the ASM group be categorized to the No-ASM group if follow-up had been extended? I don't understand.

Response: We are sorry for any confusion. To avoid reader confusion, we have removed the sentences in question.

Reviewer 2 Report

This is an interesting study, but the results should be considered as preliminary because the low sample size as a main limitation. Considering this issue, it is possible that lesion location should be different between the two study groups in larger samples. The authors should discuss on the strengths of their study.  

Author Response

Thank you very much for reviewing our manuscript. We greatly appreciate the advice provided.

This is an interesting study, but the results should be considered as preliminary because the low sample size as a main limitation. Considering this issue, it is possible that lesion location should be different between the two study groups in larger samples. The authors should discuss on the strengths of their study.  

Response: We agree with this comment. We have changed the conclusions provided in the Abstract and main body, and have mentioned the small number of patients as a limitation in the revised version of the Discussion section.
